# Divergence-Free Copy-Paste For Fluid Animation Using Stokes Interpolation

Category: Research

## ABSTRACT

Traditional grid-based fluid simulation is often difficult to control and costly to perform. Therefore, the ability to reuse previously computed simulation data is a tantalizing idea that would have significant benefits to artists and end-users of fluid animation tools. We introduce a remarkably simple yet effective *copy-and-paste* methodology for fluid animation that allows direct reuse of existing simulation data by smoothly combining two existing simulation data sets together. The method makes use of a steady Stokes solver to determine an appropriate transition velocity field across a blend region between an existing simulated outer target domain and a region of flow copied from a source simulation. While prior work suffers from non-divergence and associated compression artifacts, our approach always yields divergence-free velocity fields, yet also ensures an optimally smooth blend between the two input flows.

## 1 INTRODUCTION

The ubiquitous copy-paste metaphor from text and image processing tools is popular because it is conceptually simple and significantly reduces the need for redundant effort on the part of the user. A copy-paste tool for fluid simulation could offer similar benefits while reducing the total computational effort expended to achieve a desired results through the reuse of existing simulation data. This paper proposes exactly such a scheme.

The control of fluids has long been a subject of interest in computer animation: typical strategies that have been explored include space-time optimization (e.g., [18, 35]), space-time interpolation (e.g., [26, 33]), and approaches that involve the application of some combination of user-designed forces, constraints, or boundary conditions (e.g., [19, 24, 32, 34, 36, 37]). Because the last of these families is typically the least expensive and offers the most direct control, it has generally been the most effective and widely used in practice. Our method also falls into this category.

Inspired by Poisson image editing [23], recent work by Sato et al. [28] hints at the potential power of a copy-paste metaphor for fluids. Unfortunately, their approach suffers from problematic non-zero divergence artifacts at the boundary of pasted regions, which depend heavily on the choice of input fields. We therefore introduce a new approach that provides natural blends between source and target regions yet is relatively simple to set up, requires solving only a standard Stokes problem over a narrow blend region at each time step, and always produces divergence-free vector fields.

## 2 RELATED WORK

### 2.1 Controlling fluid animation

Artistic control of fluid flows has been a subject of interest from the earliest days of three-dimensional fluid animation research. Foster and Metaxas [11] proposed a variety of basic control mechanism through imposition of initial or boundary values on quantities such as velocity, pressure and surface tension. A wide range of subsequent methods have been proposed to enable various control methods, which we review below.

One quite common approach is to apply forces or optimization approaches to encourage a simulation to hit particular target keyframes for the density or shape of smoke or liquid [8, 18, 22, 29, 34, 35]. Another strategy makes use of multiple scales or frequencies, using a precomputed or procedural flow to describe the low-resolution motion and allowing a new physical simulation to add in high-frequency details [10, 19–21, 34].

Other approaches aim to work more directly on the fluid geometry, rather than the velocity field. For example new editing metaphors have been proposed, such as space-time fluid sculpting [17] and fluid-carving [9]; the latter is conceptually similar to seam-carving from image/video editing [1]. Another direct geometric approach seeks to directly interpolate the global fluid shape and motion [26, 33]. These strategies generally require the overall simulation to already be relatively close to the desired target behavior.

Approaches that rely on the direction application of velocity boundary conditions on the fluid flow are similar to ours in some respects [24, 25, 32, 36]. Often these have been used to cause liquid to follow a target motion or character, with varying degrees of "looseness" allowed in order to retain a fluid-like effect. They have not been used to combine existing simulations.

Another useful task in liquid animation is to insert a localized 3D dynamic liquid simulation, such as the region around a ship or swimming character, into a much larger surrounding procedural ocean or similar model. This has been achieved through the use of non-reflecting boundary conditions [6, 30]. These approaches focus on simulating the interior surface region of a liquid and smoothly damping out the surface flow to match the prescribed exterior model. This contrasts with our copy and paste problem, where both the interior and exterior are presimulated flows that must be combined together.

The closest method to ours is of course that of Sato et al. [28], who first proposed the copy-paste fluid problem. Their work also begins from the Dirichlet energy; however, through an ad hoc substitution of the input field's curl, they arrive at a new energy that minimizes the squared divergence of the velocity field plus the *difference* of the curl of the output and input vector fields. This formulation penalizes divergence rather than constraining it to be zero, and this likely accounts for the presence of undesired erroneous divergence in their results. By contrast, our approach is always strictly divergence-free.

### 2.2 Stokes flow in computer graphics

Steady (time-independent) Stokes flow is an approximation that is appropriate when momentum is effectively negligible, as indicated by a low Reynolds number. In computer graphics this approximation has been used in the context of paint simulation [4] and for design of fluidic devices [7]. The unsteady (time-dependent) variant has also been used as a substep within a more general Navier-Stokes simulator [16] for Newtonian fluids. Closest to our work is that of Bhattacharya et al. [5] who use the Stokes equations to fill a volume with smooth velocities, as an alternative to simple velocity extrapolation or potential flow approximations; they then use the generated field as a force to influence a liquid simulation. Their approach is shown to maintain rotational motion better than those existing alternative interpolants. We expand on this idea to address the copy-paste problem.

## 3 METHOD

Our method takes as input the per-timestep vector field data for two complete grid-based incompressible fluid simulations (denoted *source* and *target*), along with geometry information dividing the

domain of the final animation into an inner source region $\Omega_s$, an outer target region $\Omega_t$, and a blending region $\Omega_b$. In the final time-varying vector field to be assembled, the data in $\Omega_s$ and $\Omega_t$ are simply replayed from the inputs; the central task we must solve is to generate a "natural" vector field for the blend region $\Omega_b$ in between for all time steps.

We would like the vector field we generate in the blend region to possess a few key characteristics. First, the velocities at the boundaries of the blend region (on either $\Gamma_t = \Omega_t \cap \Omega_b = \partial \Omega_t$ or $\Gamma_s = \Omega_s \cap \Omega_b = \partial \Omega_s$) should exactly match the velocities of the corresponding input field — this is essentially the familiar no-slip boundary condition often used for kinematic solids or prescribed inflows/outflows in Newtonian fluids. Second, the vector field should be relatively smooth, since our objective is essentially a special kind of velocity *interpolant*. With only these two stipulations, a very natural choice is *harmonic interpolation* [15]. As suggested by Sato et al. [28], this can be expressed as minimizing the Dirichlet energy:

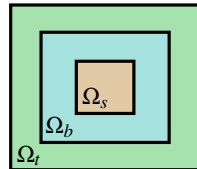

$$\underset{\mathbf{u}_b}{\text{argmin}} \iiint_{\Omega_b} \|\nabla \mathbf{u_b}\|^2 \qquad (1)$$
$$\text{subject to } \mathbf{u}_b = \mathbf{u}_s \text{ on } \Gamma_s,$$
$$\mathbf{u}_b = \mathbf{u}_t \text{ on } \Gamma_t.$$

The minimizer satisfies $\nabla \cdot \nabla \mathbf{u}_b = 0$, i.e. a componentwise Laplace equation on the velocity. (From here on we diverge from Sato et al. who proceed instead to manipulate the Dirichlet energy into a form that yields a vector Poisson equation.)

The Dirichlet energy alone is clearly insufficient, because it will prioritize smoothness at the cost of introducing divergence. Because we have assumed an incompressible flow model for our input (and desired output), the velocities in the blend region should not create or destroy material. A natural solution would be to simply apply a standard pressure projection as a post-process to convert the harmonic velocity field above to be incompressible. Unfortunately, this can cause the velocity field to deviate significantly from the harmonic input. Moreover, as we show in Section 5, pressure projection enforces only a *free-slip* condition (no-normal-flow), which allows objectionable tangential velocity discontinuities at the blend region's boundaries to be introduced.

We instead simultaneously combine the divergence-free stipulation with harmonic interpolation through the following formulation:

$$\underset{\mathbf{u}_b}{\text{argmin}} \iiint_{\Omega_b} \|\nabla \mathbf{u_b}\|^2 \qquad (2)$$
$$\text{subject to } \nabla \cdot \mathbf{u}_b = 0 \text{ on } \Omega_b,$$
$$\mathbf{u}_b = \mathbf{u}_s \text{ on } \Gamma_s,$$
$$\mathbf{u}_b = \mathbf{u}_t \text{ on } \Gamma_t.$$

This optimization problem provides the smoothest velocity field that interpolates the boundary data while preserving incompressibility. If we enforce the constraint with a Lagrange multiplier $p$, the optimality conditions turn out to yield exactly the (constant viscosity) *steady Stokes equations*,

$$\nabla \cdot \nabla \mathbf{u}_b - \nabla p = 0. \qquad (3)$$
$$\nabla \cdot \mathbf{u}_b = 0, \qquad (4)$$

consistent with Helmholtz's minimum dissipation theorem [2]. We therefore refer to this construction as *Stokes interpolation*.

As noted in Section 2, we are not the first to suggest using the Stokes equations as a fluid interpolant: Bhattacharya et al. [5] first proposed steady state Stokes flow interpolation. However, our derivation and discussion above provides additional justification and insight into the variational nature of this approach. More importantly, Bhattacharya et al. did not consider the fluid cut-and-paste problem that we address in the current work.

A minor issue is that, for there to exist a valid solution, the boundary conditions must satisfy a *compatibility* condition; that is, the integrated flux across the two boundaries must be consistent with the condition of incompressibility on the blend region's interior:

$$\iiint_{\Omega_b} \nabla \cdot \mathbf{u}^{n+1} dV = 0 = \iint_{\Gamma_s} \mathbf{u}_s^{n+1} \cdot \mathbf{n} dA + \iint_{\Gamma_t} \mathbf{u}_t^{n+1} \cdot \mathbf{n} dA \quad (5)$$

Fortunately, since the input vector fields both come from simulations that are themselves incompressible, the divergence theorem ensures that both the source copied patch and the target region to be pasted over have zero net flux across their respective boundaries – hence compatibility is guaranteed.

We arrive at the following algorithm. For each timestep, extract the boundary velocities from the input source and target simulations. Perform a steady Stokes solve on $\Omega_b$ as we have described to produce $\mathbf{u}_b^{n+1}$. Finally, directly fill in the inner and outer $\Omega_s$ and $\Omega_t$ regions with velocity from the input data $\mathbf{u}_s^{n+1}$ and $\mathbf{u}_t^{n+1}$, respectively. The resulting time-varying vector field is divergence-free and offers an attractively smooth blend between source and target flows.

Note that, since the combined vector field differs significantly from both its inputs, the flow of any passive material (such as smoke density or tracer particles) must be recomputed from scratch by advection through the new field in order to yield a consistent visual result. This can usually be done efficiently and in parallel, since each (passive) particle's motion affects no other particles.

## 4 IMPLEMENTATION

While our concept is very general, our implementation assumes that all simulations are arranged on a standard staggered ("MAC") grid [14]. This provides a natural infrastructure on which to discretize the Stokes equations on the blend region, via centered finite differences. The boundary between the blend region and the surrounding source and target flow fields is assumed to lie on axis-aligned grid faces between cells (although this could potentially be generalized to irregular cut-cells if desired [3, 16]). Where needed to ensure precise no-slip velocities conditions at the exact face midpoints on voxelized boundaries of the blend region, we make use of the usual ghost fluid method [12] for the Laplace operator in (4). To solve the Stokes linear system at each step, we use the Least Squares Conjugate Gradient solver provided by the Eigen library [13], with a tolerance of $5 \times 10^{-5}$. (Other options for solving indefinite systems, such as SYMQMR or MINRES would also be appropriate [27].)

## 5 RESULTS

We now consider some illustrative scenarios to demonstrate the behaviour of our method. Most of our figures make use of passive marker particles with alternating colors in initially horizontal rows to better highlight the developing flow structure, but we strongly encourage the reader to review our supplemental video to assess the motion more fully.

Our first scenario (Figure 1) consists of a static solid disk in a vertical wind-tunnel scenario, with inflow at the top and outflow at the bottom (particles leaving the bottom boundary re-enter at the top). We wish to paste the disk and its surroundings from the source simulation into an even simpler empty vertically translating wind-tunnel target simulation. This yields a smooth divergence-free combination of the two flows, where the flow outside of the blend region is completely undisturbed. Our result necessarily differs from the source animation, since in the source animation the presence of the disk *globally* disturbed the flow; our Stokes interpolation

approach must therefore deform the flow more strongly in the blend region to compensate, yet we still achieve a visually plausible flow (Figure 2).

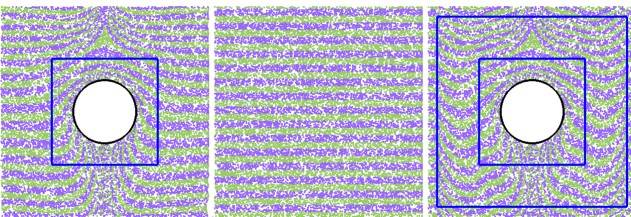

Figure 1: **Basic Setup:** Our simplest scenario involves copying the flow around a disk from its source simulation (left, with region to be copied surrounded in blue) into an obstacle-free target simulation (middle). The result of our method is a new smoothly combined flow (right). The blue lines denote the inner and outer borders of the blend region over which we apply our Stokes interpolation. (The same frame of animation is shown in all three images.)

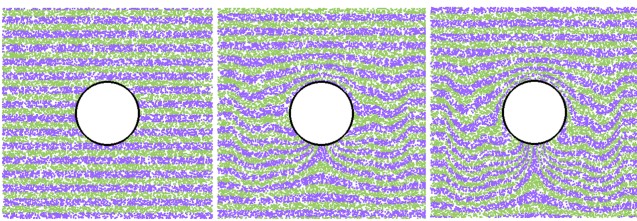

Figure 2: **Merged Flow Over Time:** A few frames of the edited animation result of our approach based on the scenario described in Figure 1.

Next, we consider our method in comparison to two other obvious alternatives, as discussed in Section 3: componentwise harmonic interpolation, and *post-projected* harmonic interpolation. Pure harmonic interpolation seems effective at first glance, but unfortunately suffers from non-negligible divergence, as shown in Figure 3.

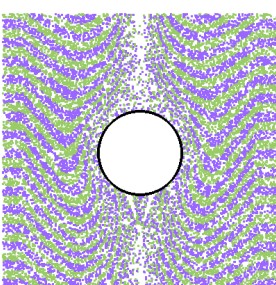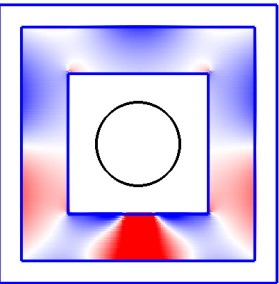

Figure 3: **Harmonic Interpolation:** Harmonic interpolation of the velocity across the blend region yields a somewhat plausible flow (left), but suffers from large divergence (right). Red indicates positive divergence, blue indicates negative divergence. The divergence gradually induces greater clumping and spreading of the particles, as seen in the middle of the left image.

A possible improvement is to post-process the harmonic result with a projection to a divergence-free state. Unfortunately, while this successfully removes divergence, the natural free-slip conditions of the pressure projection reintroduce tangential slip along the borders of the blend region leading to objectionable motion artifacts in the flow. In the wind-tunnel scenario the vertical component of

velocity suffers from discontinuities at blend region borders, leading to visible grid-aligned shearing of the flow.

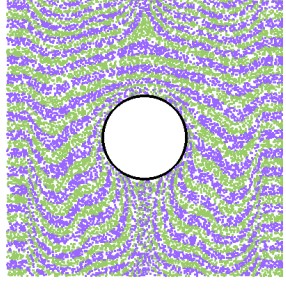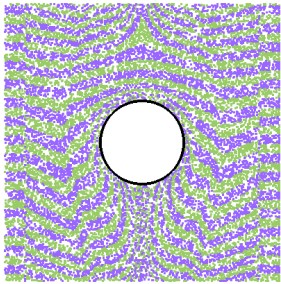

(a) Stokes Interpolation (ours)  (b) Projected Harmonic Interpolation

Figure 4: **Projected Harmonic Interpolation:** Our Stokes interpolation approach (left) yields continuous velocity fields. However, under projected harmonic interpolation (right), undesirable free-slip conditions introduce tangential discontinuities in the flow velocity at blend region borders, seen here as positional discontinuities in the rows of colored particles at far left and right.

To further stress-test our method, we consider some challenging scenarios analogous to those suggested by Sato et al. [28]. We combine flows in which the source and target differ in *direction* or *speed*. In Sato's approach, both larger speed and angle deviations lead to more severe failures of the divergence-free condition (we refer the reader to the secondary supplemental video accompanying that paper). Figure 5 shows the same test as we performed in our earlier examples, except that we have changed the ambient flow direction of the source simulation to have steadily increasing angles, including an example where the flow direction is *completely reversed*. While this leads to an increasingly unnatural look, the resulting flow field is still continuous, smooth on the blend region interior, and divergence-free independent, of this artistic decision. Similarly, Figure 6 performs a test in which the speed of the source (inner) simulation is slow or faster than the target (outer) simulation. Once again more severe speed differences lead to more unusual motions in the blend region in order to compensate. For example, when the speed ratio between source and target is 3, more elaborate interior circulation of the flow in the blend region becomes necessary to satisfy the incompressible condition. However, because the source and target are divergence-free and therefore provide compatible boundary conditions, the result is still correctly divergence-free.

A further point to note about these stress tests is that the more severe cases induce strong vortices that cause gaps to open in the flow. However, this is *not* due to divergence; rather, typical small numerical errors in particle trajectories due to interpolation and advection cause the particles to spread out from these points.

Lastly, we consider a few slightly more complex scenarios. Figure 7 shows our basic scenario again but using a rectangular obstacle instead of a disk. Figure 8 shows a scenario in which the user replaces a rectangular obstacle with a disk. Finally, in Figure 9, we paste a disk obstacle into a scene containing three rectangles, where the disk replaces the middle rectangle. Because of the additional obstacles, the flow structure is more complex. In this example, we tightened the blend region to fit more closely around the paste region. In all cases a plausible flow is constructed.

Notably, because a disk and a rectangle lead to different downstream motions in their respective wakes, a close inspection of the motion in these regions of our results reveals slightly unnatural motions, as our interpolant diligently tries to transition between two different flow structures. Fortunately, such effects are fairly subtle unless one is specifically looking for them. Ultimately, Stokes interpolation provides the best available solution under the stated

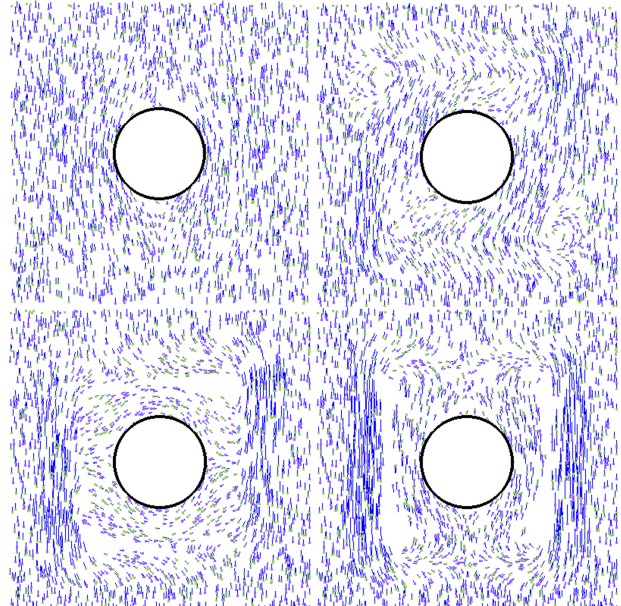

Figure 5: **Varying Angles:** In these scenes, the outer flow is vertical while the pasted inner flow from the source simulation has flow direction with a relative angle of: 0° (top-left), 45° (top-right), 90° (bottom-left), and 180° (bottom-right). In all cases, the flow remains divergence-free.

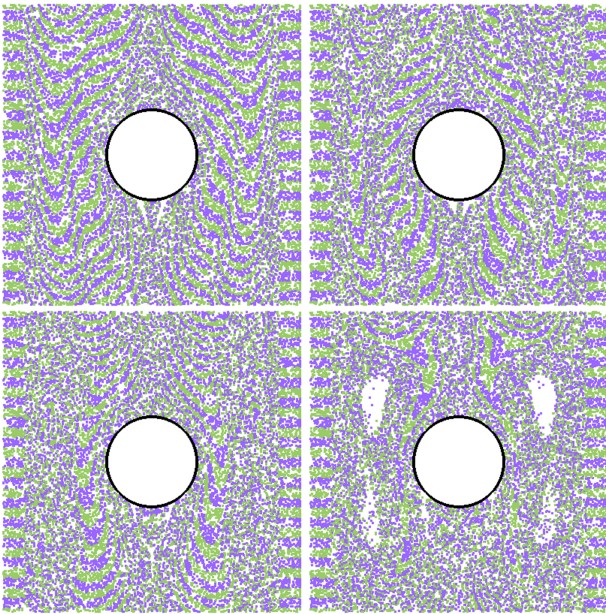

Figure 6: **Varying Speeds:** In these scenes, the outer flow has a fixed speed while the pasted inner flow from the source simulation has a speed ratio of: 1.0 (top-left), 0.75 (top-right), 1.5 (bottom-left), and 3.0 (bottom-right). In all cases, the flow remains divergence-free.

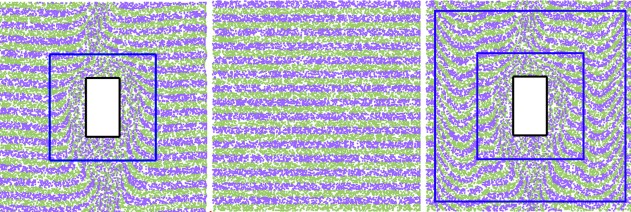

Figure 7: Pasting a rectangle into a flow. From left to right: source scene, target scene, result.

constraints (smoothness, incompressibility, interpolation of boundary values), and it is up to the user to apply their judgment regarding whether a proposed flow edit achieves the desired effect.

## 6 Conclusions and Future Work

We have presented an approach to the fluid copy-paste problem that guarantees smooth and divergence-free fields by solving a steady Stokes problem at each time step to fill in a blend region between the source and target flow regions.

Our work suggests several directions to explore in future work. First, for simplicity we assumed axis-aligned rectangles for the copy-paste region, similar to basic region-selection in image editing, but it could be useful to extend our approach to more general (lasso-type) selection regions, either in a voxelized fashion or using irregular cut-cells [3, 16] for smoother shapes. This would add greater artistic flexibility, and may render the blend-region borders less apparent.

The mathematics underlying our approach extends naturally to 3D, although providing a manageable user interface for selecting and placing time-dependent volumetric flow regions becomes more challenging. This would be interesting to explore.

Another intriguing question is whether even better behavior at blend region borders could be achieved by replacing our Dirichlet energy with a higher order energy. At present, the no-slip condition enforces matching of the velocity *value* at the boundaries, but not its *gradient*. Minimizing instead a squared Laplacian energy (see e.g., [31]), still subject to incompressibility, would lead to a bilaplacian operator on velocity. This is conceptually similar to replacing linear interpolation with cubic interpolation. While it would lead to a more challenging linear system to solve (in terms of conditioning) it may be able to offer a value- *and* gradient-matched divergence-free blend field.

Finally, a challenging unanswered question in fluid animation more broadly is to what makes a fluid motion perceptually "realistic" from a human perspective, and how much deviation from physical accuracy can safely be tolerated in visual applications. A metric of this kind could allow one to quantify more concretely whether a proposed flow edit is successful or harmful.

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
