# OpenReview forum: "Divergence-Free Copy-Paste For Fluid Animation Using Stokes Interpolation"
_graphicsinterface.org/Graphics_Interface/2021/Conference — Submitted to GI 2021_

### Official Review · AnonReviewer2 · 2021-01-06
**Difficult to see method advantage in examples**

**Rating:** 4
**Confidence:** 3

**Review:**

The paper presents a divergence-free method to blend two fluid simulations. This is an improvement over Sato et al which has no divergence-free guarantee. But the exact math is not new. On top of that, the results don't seem to have much visual improvement. There is still visuallly unnatural deformation and the field near the boundary between target region (outer) and blend region (middle) still has sharp changes. The animation created by harmonic interpolation seems more natural (about 1:00 in the video).

I think the major problem is that the examples don't show how good the method is. It would improve the paper if there are more comparisons that can show the problems of the prior work's method (e.g discontinuities, divergence) while the authors' method is visually better.

Some comments about the figures:

- Figure 1 right and Figure 7 right have the outer region animation changed (strips not straight). But the authors' method is supposed to not alter the target animation outside the boundary of blend region.

- Figure 2 left is not a frame of the resulting animation.

- In Figure 4, it is difficult to notice any smoothness differences between the two methods. Both have some very sharp changes in the field. The authors can highlight the discontinuities in the right image.

---

### Official Review · AnonReviewer1 · 2021-01-13
**Results are limited in 2D, but potentially valuable**

**Rating:** 6
**Confidence:** 3

**Review:**

This paper proposes a new fluid-interpolation scheme with a divergence-free constraint. The gap between the two input flows are smoothly interpolated using a devised vector laplacian.

The previous work did not enforce divergence-free constraints, and the idea of enforcing "divergence-free" constraint at the boundary also satisfies the internal divergence-free is both conceptually and mathematically interesting. Such an idea and the implementation are exciting, and I am in favor of accepting.

At the moment, the results are only limited to 2D, and I would like to see some 3D examples. I encourage the authors to provide at least one 3D example if the paper is accepted in the final publication, but this is not mandatory.

---

### Official Review · AnonReviewer3 · 2021-01-13
**Novel application of stokes equations for fluid copy and paste, but comparison and results need work**

**Rating:** 5
**Confidence:** 3

**Review:**

This paper proposes a method for "copy and paste" editing of fluid simulations that combines regions from two different simulations by blending the vector fields across a boundary region using steady state Stokes interpolation and assuring a divergence-free flow.

The proposed approach seems to be a novel application of the work of Bhatacharya et al (2012). So while I find the contribution here to be modest, the problem is nicely motivated by providing insight into artefacts created by cut and paste editing, mainly i) non smooth and ii) divergent flows. These artefacts are resolved by the proposed approach by assuring that the boundary regions have smooth flows *and* are divergence-free by solving a constrained least squares problem.  As noted in the paper, the input vector fields already exhibit properties of physical fluids.

The related work discussion and background in this paper is excellent, and the authors have done a good job of situating their work relative to others in the field. However, not being an expert in fluid simulation, I would have appreciated a bit more detail about the cfd formulation used in this work, if only to make the paper more self contained and to define specific terms used throughout the paper.

My issue with this work, and it is a pretty major one, is that the results presented in the paper and supplementary video make it difficult to properly evaluate the approach.

I understand that the two-tone tracer particles used in the fluid simulation may have been chosen to highlight artefacts, but in many cases the patterns produced are quite noisy and it is difficult to identify details, e.g. in Figure 4, flow velocity discontinuities are not clearly visible.  Perhaps larger figures would help.  Rendering smoke density may also make it easier to compare to results produced by other work, such as Sato et al 2018. Specifically, it would have been nice to see a comparison to tests performed in the supplementary video of Sato et al, in which they evaluated their approach for different directions and magnitudes of velocity fields.  I think the approach in this paper would perform better.

However, in order to provide convincing evidence about the utility of the approach, a 3D implementation would best demonstrate how the method handles complex examples.  User interface challenges aside, it should be straightforward to extend the approach to 3D (as noted in Section 6) and demonstrate the technique with at least one compelling 3D example.

Also, there are also no timing results provided in the paper, but they would be helpful to know and specifically about the overhead compared to projected harmonic interpolation. However, I think timing information would only be meaningful for 3D simulations.

---

### Meta-Review · Area_Chair1 · 2021-01-16

**Recommendation:** Reject
**Confidence:** 4

**Metareview:**

Dear Authors,

As a Area Chair on this paper, I have collected three reviews and examined all of them ( I am one of them). To summarize, all the reviewers pointed out that the quality of the paper needs several improvements, which include clarity of mathematical formulation, extension to 3D examples, comparisons and timings. In reading the reviews, I also find that two reviewer finds that the idea of paper is novel and one reviewer finds that the paper is acceptable.

Overall, I recommend rejection because only one of us recommends (weak) accept, and two of us believe that the paper still does not meet the bar to be published in the venue. However, as I mentioned in the beginning, this recommendation is mainly due to the lack of missing materials (clearer formulation, comparisons, timings, 3D extensions, etc.). Therefore, as a Area Chair, I would like to encourage authors to keep polishing this work and possibly re-submit to some graphics-related journal/conference.

Sincerely,
Area Chair

---

### Decision · Program_Chairs · 2021-01-16

Reject